# Associations between Systemic Immune-Inflammation Index and Diabetes Mellitus Secondary to Pancreatic Ductal Adenocarcinoma

**DOI:** 10.3390/jcm12030756

**Published:** 2023-01-18

**Authors:** Guanhua Chen, Chunlu Tan, Xubao Liu, Xing Wang, Qingquan Tan, Yonghua Chen

**Affiliations:** Department of Pancreatic Surgery, West China Hospital of Sichuan University, Chengdu 610041, China

**Keywords:** pancreatic ductal adenocarcinoma, diabetes mellitus, inflammation, systemic immune-inflammation index

## Abstract

Background: There is a high prevalence of diabetes mellitus (DM) in patients with pancreatic ductal adenocarcinoma (PDAC). An inflammatory response is considered as a potential mechanism involved in the process. The systemic immune-inflammation (SII) index is an integrated and novel inflammatory indicator developed in recent years. The purpose of this study was to determine the relationship between the SII and DM secondary to PDAC. Method: Patients with a confirmed diagnosis of PDAC were analyzed in this cross-sectional study. Anthropometric measures, glucose-related data (including fasting glucose, 2 h OGTT, glycated hemoglobin, fasting insulin, and fasting c-peptide), tumor characteristics (tumor volumes, location and stages), and the periphery blood inflammatory index (white blood cell count, neutrophil-to-lymphocyte ratio, platelet-to-lymphocyte ratio, and SII) were recorded. The inflammation index was analyzed for its association with glucose-related parameters. Multivariable logistic regression analysis was used to analyze the association between SII levels and DM secondary to PDAC. Results: Blood cell results showed that the white blood cell count, neutrophils, lymphocytes, monocytes, platelets, the neutrophil-to-lymphocyte ratio, and platelet-to-lymphocyte ratio were higher in patients with diabetes. It was worth noting that SII significantly increased in patients with diabetes secondary to PDAC (4.41 vs. 3.19, *p* < 0.0001). Multivariable logistic regression analysis showed that SII (OR: 2.024, 95%CI: 1.297, 3.157, *p* = 0.002) and age (OR: 1.043, 95%CI: 1.01, 1.077, *p* = 0.011) were the risk factors for DM secondary to PDAC after adjusting for covariates. According to Spearmen correlation analysis, SII was positively correlated with fasting glucose (r = 0.345, *p* < 0.0001), 2 h OGTT (r = 0.383, *p* < 0.0001), HbA1c (r = 0.211, *p* = 0.005), fasting insulin (r = 0.435, *p* < 0.0001), fasting C-peptide (r = 0.420, *p* < 0.0001), and HOMA2-IR (r = 0.491, *p* < 0.0001). Conclusions: In conclusion, SII is significantly increased among patients with DM secondary to PDAC and is associated with the DM in patients with PDAC (OR: 2.382, 95% CI: 1.157, 4.903, *p* = 0.019). Additionally, SII is significantly correlated with insulin resistance. We are the first to investigate the relationship between SII and diabetes secondary to PDAC and further confirm the role of an inflammatory response in this process. More studies need to be designed to clarify how inflammatory responses participate.

## 1. Introduction

Globally, the alarmingly escalation of people with diabetes mellitus (DM) diagnoses has been accompanied by increasing rates of some type of cancers, especially pancreatic ductal adenocarcinoma (PDAC) [1], which has led to a hypothesis of a potential direct relationship between the two diseases. Diabetes mellitus and PDAC share common risk factors (e.g., old age, obesity, and family history of diabetes), and long-standing type 2 diabetes mellitus represents a recognized risk for carcinogenesis [2]. New-onset DM can be a manifestation of PDAC [3]. Recently, more and more compelling epidemiological and clinical evidence now supports the concept that diabetes in the context of pancreatic cancer is a paraneoplastic phenomenon which is typically named as pancreatic diabetes or type 3c diabetes mellitus (T3cDM). In this form of diabetes mellitus, fibrosis, sclerosis of pancreatic tissue, and inflammation are commonly considered as the underlying mechanisms [4].

Prior research has shown that inflammation has been implicated as the potential mechanism responsible for the occurrence of diabetes and its complications in the general population. The establishment of DM occurs in most, if not all, cases in the presence of chronic inflammation and is associated with the secretion of large amounts of pro-inflammatory cytokines such as IL-1, IL-6, CRP, and TNF-a [5]. Recently, several studies have proved the neutrophil-to-lymphocyte ratio (NLR), monocyte-to-lymphocyte ratio (MLR), and platelet-to-lymphocyte ratio (PLR) as inflammation biomarkers in the context of diabetes [6,7,8]. A systemic immune-inflammation index (SII) has been considered as a good, stable index that reflects both local immune responses and systemic inflammation. It integrates three types of inflammatory cells, including platelet, neutrophil, and lymphocyte, and has been proposed as an indicator for systemic inflammation, which is a risk factor for major cardiovascular events and a predictor of cancer outcomes [9,10,11,12]. However, the detail of the mechanisms of T3cDM due to PDAC has not been elucidated. Thus, in our study, we aimed to investigate the associations between SII and diabetes mellitus secondary to PDAC.

## 2. Subjects, Materials and Methods

### 2.1. Study Design and Participants

This was a cross-sectional study conducted at the Department of Pancreatic Surgery, West China Hospital. We aimed to explore the potential relationship between SII and diabetes mellitus secondary to PDAC in patients. The patients hospitalized in the Department of Pancreatic Surgery, West China Hospital, from August 2018 to December 2020 were recruited. The eligible criteria were as follows: (1) patients at age 18 and above, (2) patients with pathologically diagnosed PDAC, (3) newly diagnosed diabetes within 12 months of confirmed PDAC, and (4) patients with a valid consent form. Exclusion criteria were: (1) patients with known diabetes for 1 year or longer (n = 7); (2) patients with some clinical context that may interfere with the results, including acute inflammatory disease (n = 38); immune disorders (n = 16); and hepatic or kidney failure (n = 2). A total of 2 patients were missing data (n = 7). Overall, 291 patients were recruited, about 70 patients were excluded, and 221 patients were finally analyzed in our study. All study participants or their legal guardian provided informed written consent for personal and medical data collection prior to study enrolment. We were reviewed and approved by the Medical Ethics Committee of West China Hospital at Sichuan University.

### 2.2. Measurements

Information on demographics including age, gender, height, and weight was collected from medical data. The body mass index (BMI) was calculated as the weight in kilograms divided by the height in meters squared. Blood samples were collected from all participants before surgery and were used for the measurements of complete blood count, glycated hemoglobin (HbA1c), fasting insulin, fasting C-peptide, and plasma glucose. After a 10 h overnight fast, an oral glucose tolerance test (OGTT) was performed using 75 g of anhydrous glucose dissolved in 3 dL of water, and we collected plasma glucose at 0 min and 120 min. The plasma glucose was tested using the glucose oxidase method (a cobas 8000 analyzer, Roche Diagnostics, Basel, Switzerland). HbA1c was measured in venous blood samples by high performance liquid chromatography (BioRad VARIANT II Hemoglobin Analyzer). Insulin resistance and beta cell function were estimated using the HOMA2 model (available from www.OCDEM.ox.ac.uk on 2 January 2021) according to the recommendations for its appropriate use. A complete blood count was recorded using an automatic blood cell analyzer (Cysmex, USA), and we collected neutrophil count, lymphocyte count, and platelet count. The systemic immune-inflammation index was calculated as neutrophil × lymphocyte/platelet. In addition, two independent pathologists performed pathological examination on postoperative specimens. The 8th AJCC stage adapting to the patients was divided into two groups: I–II and III–IV [13]. Other blood parameters including lipids, creatine, albumin, and alanine aminotransferase were also collected.

### 2.3. Definition of DM

We used the 2020 American Diabetes Association guidelines to categorize test results at two levels: normal and diabetes mellitus. For this analysis, we used OGTT data at 2 h only. Based on OGTT, a 2 h glucose level ≥11.1 mmol/L indicated diabetes, and ≤11.1 mmol/L indicated non-DM [14].

### 2.4. Statistical

Continuous variables are expressed as mean± standard deviation (SD) or median and interquartile range (IQR) as appropriate, and categorical variables as percentages (%). Variables with a non-normal distribution were compared with the Mann–Whitney U test. Categorical variables were compared using the chi-square test. Using multivariate logistic regression analysis, odds ratios (ORs) and 95% confidence intervals (CIs) were calculated for the association between SII levels and diabetes. Spearman correlation analysis was assessed between inflammatory biomarkers and clinical parameters. All statistical analyses were carried out using the Statistical Package for Social Sciences (SPSS) Version 26. A *p*-value of 0.05 was considered statistically significant.

## 3. Results

### 3.1. Subject Characteristics

There were 221 patients with pathologically confirmed PDAC in our study. The median age of patients was 60 (50, 67), and 59.2% (n = 131) of them were male. Furthermore, 35.8% (n = 77) of them were diagnosed with new-onset diabetes by OGTT within 12 months of diagnosis of PDAC, including 17 (22.1%) with a confirmed diagnosis of diabetes mellitus outside the admission with the median time of 5.0 (2.0, 7.0) months. The characteristics according to diabetic status are presented in Table 1. Compared to patients without DM, those with diabetes tended to be elder (62 vs. 58, *p* = 0.031) and male (59.3% vs. 40.7%, *p* = 0.007) in our study. Larger tumor volumes (7.94 cm^3^ vs. 2.55 cm^3^, *p* < 0.0001) and more advanced tumor stages (III or IV stage 34.8% vs. 12.5%, *p* = 0.006) were also observed in those with diabetes. Blood cell results showed that WBCs, neutrophils, lymphocytes, monocytes, platelets, NLR, and PLR were higher in patients with diabetes. It was worth noting that SII significantly increased in patients with diabetes secondary to PDAC (4.41 vs. 3.19, *p* < 0.0001).

### 3.2. Spearman Correlation Analysis between Periphery Blood Indexes and Glucose-Related Parameters

Table 2 shows the correlation between periphery blood inflammatory indexes and the glucose-related index in our study. In general, those periphery blood inflammatory indexes were significantly correlated with the glucose-related index to a degree. No significance was observed between lymphocyte- and glucose-related indexes. We also found that no significance existed between HOMA2-B and any inflammatory indexes. It is noteworthy that SII was positively correlated with fasting glucose (r = 0.345, *p* < 0.0001), 2 h OGTT (r = 0.383, *p* < 0.0001), HbA1c (r = 0.211, *p* = 0.005), fasting insulin (r = 0.435, *p* < 0.0001), fasting C-peptide (r = 0.420, *p* < 0.0001), and HOMA2-IR (r = 0.491, *p* < 0.0001). Figure 1 shows the Spearman correlation analysis between periphery blood indexes and glucose-related parameters. Additionally, we observed that some inflammatory indexes were positive with tumor volumes except for lymphocyte or platelet. The r correlation coefficient was megascopic when the NLR and SII were set as variables.

### 3.3. Risk Factors for Diabetes Secondary to PDAC

In order to account for multiple risk factors concomitantly contributing to the occurrence of diabetes, logistic regression analysis was used (Table 3). In the univariate analysis, age (OR: 1.027, 95%CI: 1.002, 1.053, *p* = 0.035), gender (OR: 2.237, 95%CI: 1.236, 4.047, *p* = 0.008), tumor volumes (OR: 1.040, 95%CI: 1.011, 1.070, *p* = 0.006), tumor stage (OR: 2.625, 95%CI: 1.299, 5.307, *p* = 0.007), alanine aminotransferase (OR: 1.018, 95%CI: 1.001, 1.035, *p* = 0.040), WBC (OR: 1.395, 95%CI: 1.165, 1.668, *p* < 0.0001), and platelet (OR: 1.011, 95%CI: 1.005, 1.017, *p* < 0.0001) were associated with a higher risk of DM. After entering the above variables into multiple logistic analysis, the results showed that age (OR: 1.043, 95%CI: 1.01, 1.077, *p* = 0.011) and SII (OR: 2.024, 95%CI: 1.297, 3.157, *p* = 0.002) were both positively associated with DM among patients with PDAC.

## 4. Discussion

The association between diabetes mellitus and PDAC has been noted for more than 200 years. There is a threefold increase in diabetes prevalence among PDAC patients compared to common cancers [15]. Increased prevalence of diabetes and hyperglycemia in patients with PDAC of up to 80% has been reported in previous studies [16,17]. Some studies have indicated inflammation plays an important role in diabetes and some pro-inflammation cytokines, and this inflammation is correlated with the onset of diabetes. However, no study was accessible on the role of inflammation in diabetes secondary to PDAC. In our study, we found that SII was significantly higher in patients with diabetes secondary to PDAC. Age, gender, tumor volumes, the AJCC stage, ALT, WBC, NLR, PLR, and SII were associated with diabetes using univariate analysis. After multiple logistic analyses, age and SII were two independent risk factors for diabetes secondary to PDAC. To our knowledge, this is the first study to investigate the relationship between SII and DM secondary to PDAC.

The presence of hyperglycemia was considered as a result of the failure to maintain adequate pancreatic β-cell function to compensate for a decline in insulin sensitivity [18]. An inflammatory response has been implicated in the pathophysiology of glucose disorder [19,20]. Many lines of evidence have shown that the chronic activation of intracellular proinflammatory pathways within insulin target cells can lead to insulin resistance [20]. Clinical data also show that systemic levels of pro-inflammatory cytokines, including tumor necrosis factor (TNF)-a, interleukin (IL)-1b, IL-6, and CRP are elevated in patients with both type 1 and type 2 diabetes [21,22,23]. Moreover, these cytokines and inflammatory mediators, particularly TNF-α, monocyte chemotactic protein-1 (MCP-1), CRP, and interleukins, are considered as the potential cause of insulin resistance or impaired B cell function [24].

PDAC-induced diabetes patients have previously been shown to have increased insulin resistance and impaired B cell function [25]. The precise mechanisms have remained an active area of investigation; however, data indicate that inflammatory process may play an important role. Increased cytokine release caused by a tumor macroenvironment will cause local/systemic inflammation and lead to insulin resistance, B cell dysfunction, and insulin autonomy [26,27]. Suresh Charithe et al. indicated that PC patients’ endocrine pancreas exhibits distinct pathological characteristics such as a reduction in β- and α–cell fractional areas, reduced islet size, and substantial isle decline, and inflammatory responses are possibly responsible for the change [28]. In addition, according to a proteomic study by Dr Wang, they found the upregulation of S100-A9, expressed predominantly in immune cells, such as granulocytes, monocytes, myeloid-derived suppressor cells, and other immature cells of the myeloid lineage, in pancreatic ductal adenocarcinoma tissues with diabetes compared with those without diabetes [29]. Matrix metalloproteinase-9 (MMP9) was upregulated by macrophages in response to TNFα and interleukin 1β enriched in the tumor microenvironment DM secondary to PDAC. Thus, this further confirmed that diabetes induced by PDAC was a paraneoplastic phenomenon and inflammation plays an important role in the process [30].

The total white blood cell count, which reflects the sub-inflammation, has been considered to be related to insulin resistance and the occurrence of diabetes; however, it fails to account for the different kinetics of leukocyte subsets [31]. Previous studies suggested that activated neutrophils are first-line immune cells involved in inflammation, and type 2 diabetes is associated with modest increases in neutrophil counts [32]. In our study, the neutrophil count was higher in patients with diabetes secondary to PDAC. Moreover, a significant correlation was shown that neutrophils were significant for glucose-related indexes (except for HOMA2-B) and tumor volumes.

As well as being involved in coagulation, platelets contain a lot of pro-inflammatory molecules that regulate immune and inflammatory responses. Some studies indicated that platelets are characterized to be hyperactive with increased activation, adhesion, and aggregation due to the dysregulation of several signaling pathways in T2DM patients [33], among which low-grade vascular inflammation mediated by nuclear factor-κB (NF-κB) was found in diabetes [34,35]. In diabetes, hyperglycemia and insulin resistance may affect the endothelial function and change the function or platelet count. In our study, the correlation between platelets and insulin resistance was observed. No significance was found between platelets and glucose; however, there was a significant correlation between platelets and 2 h OGTT, which needs further investigation.

There was inconsistency in observation of lymphocytes in previous studies. In our study, although there was a decreased lymphocyte count in diabetics with PDAC, no significance exists. In the adaptive immune system, lymphocytes play a key role in protecting the body. There has been a documented increase in apoptosis in rats and patients with diabetes, as well as elevated levels of oxidative DNA damage in peripheral blood lymphocytes. This difference may be due to the limited sample size or the effect of PDAC. In addition, we found an elevated monocyte count in diabetes in accordance with previous studies. Increased monocyte presence in the tumor microenvironment or in circulation has been implicated in angiogenesis, tumor growth, and poor prognosis for cancer patients [36]. Some studies also found increased monocytes in patients with T2DM [37]; however, the relationship between the two needs to be explored.

During the past few decades, integrative cellular immune inflammation markers have emerged in clinical settings. The neutrophil-to-lymphocyte ratio, platelet-to-lymphocyte ratio, and monocyte-to-lymphocyte were usually used as efficient biomarkers for predicting the prognosis for types of cancer including PDAC. Further, there exists, to some extent, relationships between those inflammatory indexes and diabetes [38,39]. However, results from different studies vary. Currently, and based on circulating neutrophil counts, lymphocyte counts, and platelet counts, SII is a novel and integrated inflammatory biomarker and is considered as a readily detectable biomarker for systemic inflammatory activity [40]. The SII index includes three significant blood cell components linked to metabolic disorders, in particular neutrophils, platelets, and lymphocytes, and its close relationship with insulin resistance and in DM patients emphasizes even more the comprehensive role of inflammation and inflammatory indexes. In our study, we found the values of SII are higher in diabetes among PDAC patients. In addition, SII correlated with the glucose-related index and tumor volumes. Previous studies on SII focused more on its prognosis for solid tumors and the risk of cardiovascular diseases, while there were few studies on its metabolism, especially in diabetes secondary to pancreatic cancer, and its mechanism needed further research.

There were some limitations in our study. Firstly, this was a cross-sectional study and did not allow us to investigate causal associations between SII and diabetes secondary to PDAC. Furthermore, there is no universal definition of new-onset diabetes as of now. Our study participants were only those with PDAC, but the study did not include those with type 2 diabetes or healthy people. Additionally, it is difficult to accurately distinguish type 2 diabetes mellitus from T3cDM, so the mixed effect may exist. Indeed, the exact relationship between SII and diabetes secondary to PDAC as well as the mechanism effect remains to be studied.

## 5. Conclusions

In conclusion, SII is significantly increased among patients with DM secondary to PDAC and is positively associated with the risk of DM in patients with PDAC (OR: 2.382 95% CI: 1.157, 4.903, *p* = 0.019). Additionally, SII is significantly correlated with insulin resistance. We are the first to investigate the relationship between SII and diabetes secondary to PDAC and further confirmed the role of an inflammatory response in this process. More studies need to be designed to clarify how inflammatory responses participated.

## Figures and Tables

**Figure 1 jcm-12-00756-f001:**
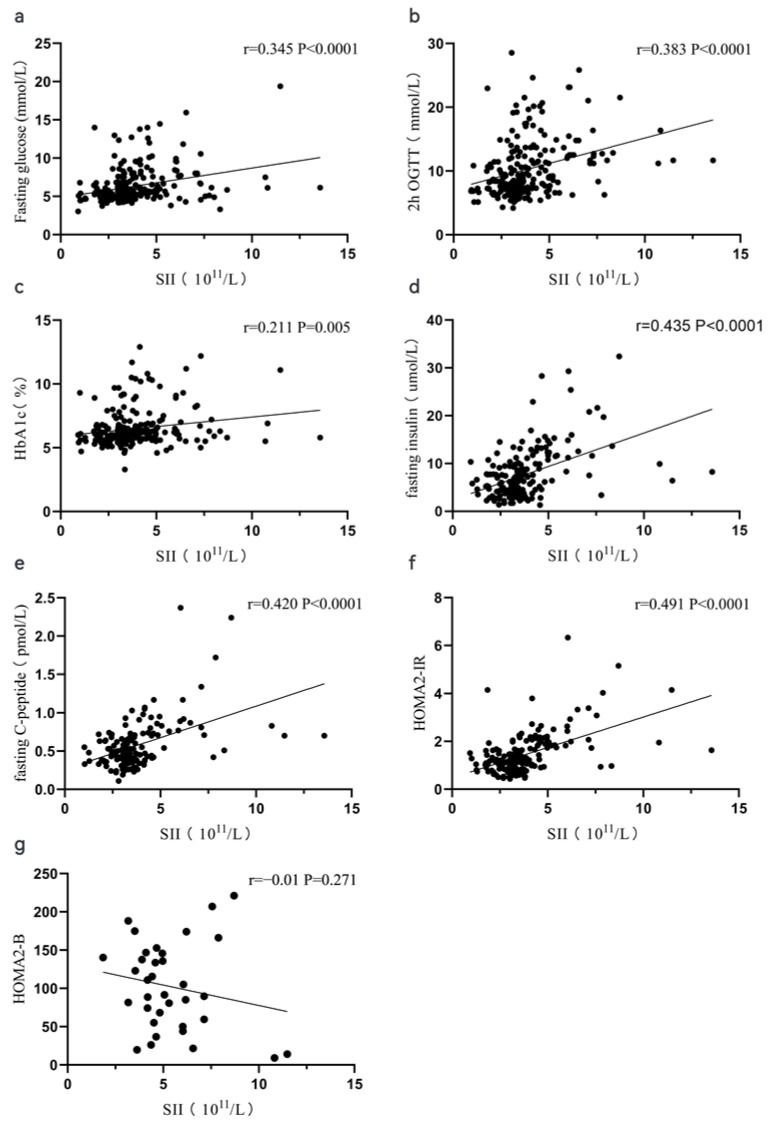
A scatter plot of the Spearman correlation analysis between SII and the glucose-related index. The correlation between SII and fasting glucose (**a**), 2 h OGTT (**b**), HbA1c (**c**), fasting insulin (**d**), fasting C-peptide (**e**), HOMA2-IR (**f**), and HOMA2-B (**g**).

**Table 1 jcm-12-00756-t001:** Patient characteristics according to diabetes mellitus (DM) status.

Variable	All (n = 221)	Non-DM (n = 144)	DM (n = 77)	*p*
Age (year)	60 (50, 67)	57 (49, 66)	62 (52, 68)	0.031
Gender (Male, %)	131	76 (52.8)	55 (59.3)	0.007
Body mass index (kg/m^2^)	23.63 (21.15, 26.04)	23.11 (21.15, 25.51)	26.34 (21.78, 26.25)	0.071
Fasting glucose (mmol/L)	5.51 (4.91, 6.85)	5.18 (4.77, 5.65)	7.57 (5.87, 9.55)	<0.0001
2 h OGTT (mmol/L)	8.70 (6.98, 12.41)	7.40 (6.53, 8.55)	13.70 (12.30, 17.44)	<0.0001
HbA1c (%)	6.1 (5.6, 6.7)	5.9 (5.6, 6.2)	6.8 (6.2, 8.6)	<0.0001
Fasting insulin (umol/L) ^a^	6.21 (3.62, 9.96)	5.83 (3.60, 9.47)	7.50 (3.58, 12.08)	0.157
Fasting C-peptide (pmol/L) ^b^	0.49 (0.38, 0.71)	0.47 (0.38, 0.67)	0.53 (0.39, 0.77)	0.250
HOMA2-B ^a^	87.30 ± 41.94	95.89 ± 36.17	68.91 ± 47.53	<0.0001
HOMA2-IR ^a^	1.21 (0.88, 1.82)	1.12 (0.84, 1.61)	1.43 (1.03, 2.10)	0.003
Tumor volume (cm^3^)	4.00 (1.52, 10.40)	2.55 (1.00, 8.75)	7.92 (2.88, 13.68)	<0.0001
Location, (body and tail %)	101 (45.7)	65 (45.1)	36 (46.8)	0.818
AJCC stage (Ⅲ~Ⅳ, %)	39 (18)	18 (12.5)	21 (34.8)	0.006
CA19-9	18.00 (9.41, 55.00)	17.50 (9.80, 44.96)	19.00 (7.52, 77.00)	0.699
Total bilirubin (umol/L)	11.10 (8.40, 13.40)	10.80 (8.20, 13.08)	11.50 (8.80, 14.30)	0.211
Alanine aminotransferase (IU/L)	18.00 (12.00, 28.00)	18.00 (13.00, 25.00)	18.00 (11.50, 32.00)	0.371
Albumin (g/L)	43.20 (40.65, 46.10)	43.10 (40.9, 45.88)	43.40 (39.60, 46.25)	0.838
Creatinine (umol/L)	66.00 (56.00, 77.00)	63.00 (54.25, 76.75)	67.00 (58.50, 77.50)	0.073
Triglyceride (mmol/L)	1.16 (0.87, 1.49)	1.10 (0.85, 1.48)	1.21 (0.94, 1.54)	0.144
Cholesterol (mmol/L)	4.26 ± 0.95	4.31 ± 0.96	4.18 ± 0.93	0.322
High density lipoprotein (mmol/L)	1.20 (0.99, 1.52)	1.22 (1.03, 1.52)	1.15 (0.94, 1.46)	0.089
Low density lipoprotein (mmol/L)	2.48 (2.01, 2.95)	2.49 (1.99, 2.98)	2.48 (2.21, 2.94)	0.913
Hemoglobin (g/L)	132.29 ± 14.61	131.28 ± 14.59	134.17 ± 14.56	0.161
White blood cell(10^9^/L)	5.37 (4.58, 6.34)	5.12 (4.32, 5.99)	5.84 (4.96, 7.52)	<0.0001
Platelet (10^9^/L)	145.00 (120.75, 182.25)	132.50 (114.25, 168.00)	170.00 (141.00, 200.00)	<0.0001
Neutrophil (10^9^/L)	3.31 (2.73, 4.26)	2.10 (2.40, 3.90)	3.82 (2.91, 4.80)	<0.0001
Lymphocyte (10^9^/L)	1.46 (1.14, 1.76)	1.46 (1.15, 1.73)	1.48 (1.10, 1.89)	0.967
Monocyte (10^9^/L)	0.39 (0.31, 0.51)	0.37 (0.29, 0.45)	0.46 (0.34, 0.59)	<0.0001
NLR	2.41 (1.85, 3.02)	2.31 (1.81, 2.79)	2.55 (1.91, 3.59)	0.003
PLR	107.69 (83.06, 134.06)	103.62 ± 33.71	131.18 ± 62.49	<0.0001
SII (10^11^/L)	3.42 (2.96, 4.58)	3.19 (2.50, 3.71)	4.41 (3.43, 6.23)	<0.0001

^a^, a fasting insulin test was performed for 179 patients, and the value of HOMA-B and HOMA-IR were calculated. ^b^, a fasting C-peptide test was performed in 140 patients. HbA1c, glycated hemoglobin, NLR neutrophil-to-lymphocyte ratio, PLR platelet-to-lymphocyte ratio, SII, systemic immune-inflammation index. The significance level was set as *p* < 0.05 (compared to the non-DM group).

**Table 2 jcm-12-00756-t002:** Spearman correlation analysis between periphery blood indexes and glucose-related parameters.

	White Blood Cell	Neutrophil	Lymphocyte	Monocyte	Platelet	NLR	PLR	SII
Fasting glucose	0.181 **	0.215 **	−0.058	0.154 *	0.110	0.251 ***	0.150 *	0.345 ***
2 h OGTT	0.174 **	0.206 **	−0.060	0.151 *	0.210 **	0.223 ***	0.236 **	0.383 ***
Fasting insulin	0.288 ***	0.313 ***	0.015	0.161 *	0.214 **	0.211 **	0.160 *	0.435 ***
Fasting C-peptide	0.371 ***	0.382 ***	0.059	0.281 **	0.154	0.248 **	0.084	0.420 ***
HbA1c	0.178 **	0.167 *	0.059	0.197 **	0.110	0.109	0.037	0.211 **
HOMA2-IR	0.358 ***	0.409 ***	0.025	0.220 **	0.173 *	0.294 ***	0.113	0.491 ***
HOMA2-B	−0.045	0.029	−0.049	−0.06	−0.013	0.012	0.002	−0.01
Tumor volumes	0.361 ***	0.445 ***	−0.107	0.284 ***	0.039	0.479 ***	0.147 *	0.478 ***

Significance set as: * < 0.05 ** < 0.01 *** < 0.0001.

**Table 3 jcm-12-00756-t003:** ORs for diabetes mellitus by logistics analysis.

Variables	Univariate Analysis		Multiple Analysis	
	OR (95%CI)	*p*	OR (95%CI)	*p*
Age	1.027 (1.002, 1.053)	0.035	1.043 (1.01, 1.077)	0.008
Gender (male)	2.237 (1.236, 4.047)	0.008	1.743 (0.827, 3.672)	0.144
Body mass index	0.998 (0.991, 1.005)	0.527	NA	
Tumor volumes	1.040 (1.011, 1.070)	0.006	0.999 (0.957, 1.043)	0.970
Tumor location (body and tail)	1.067 (0.613, 1.859)	0.818	NA	
AJCC stage (Ⅲ~Ⅳ)	2.625 (1.299, 5.307)	0.007	2.021 (0.596, 6.853)	0.259
CA19-9	1.001 (0.999, 1.002)	0.290	NA	
Total bilirubin	1.026 (0.976, 1.079)	0.316	NA	
Alanine aminotransferase	1.018 (1.001, 1.035)	0.040	1.019 (0.998, 1.039)	0.072
Albumin	0.992 (0.942, 1.044)	0.756	NA	
Creatinine	1.013 (0.996, 1.031)	0.141	NA	
Triglyceride	1.232 (0.831, 1.827)	0.299	NA	
Cholesterol	0.862 (0.642, 1.156)	0.321	NA	
High density lipoprotein	0.600 (0.283, 1.273)	0.183	NA	
Low density lipoprotein	0.866 (0.604, 1.242)	0.435	NA	
Hemoglobin	1.014 (0.994, 1.034)	0.162	NA	
White blood cell	1.395 (1.165, 1.668)	<0.0001	1.215 (0.775, 1.906)	0.396
NLR	1.846 (1.361, 2.503)	<0.0001	0.593 (0.321, 1.093)	0.094
PLR	1.016 (1.008, 1.025)	<0.0001	1.006 (0.984, 1.029)	0.598
Systemic immune-inflammation index	2.156 (1.674, 2.776)	<0.0001	2.382 (1.157, 4.903)	0.019

NA not applied. Significance set as *p* < 0.05.

## Data Availability

Not applicable.

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
