# Peer review of "Associations between Systemic Immune-Inflammation Index and Diabetes Mellitus Secondary to Pancreatic Ductal Adenocarcinoma"

_jcm, 2023, doi:10.3390/jcm12030756_

Round 1

Reviewer 1 Report

Congratulations, I like your article. 

The paper mainly focused on the study between SII and diabetes mellitus secondary to pancreatic ductal adenocarcinoma. Because this article takes advantage of the utility of the white cell count as a simple and reliable index of inflammation in the development of diabetes due to pancreatic damage, it is relevant to study it in any disease entity, although the usefulness of this index is widely known, 

Inflammation plays an important role in diabetes; however, the onset of hyperglycemia in pancreatic cancer is usually sudden, unlike type 2 DM.  References are appropriate and tables and figures are fine.

Perhaps you should have considered including a control group of healthy subjects and another group with diabetes mellitus type 2

Author Response

Dear reviewer:

Thank you very much for your comments and suggestions. Your suggestion hit the nail on the head, which made us understand our shortcomings. We accepted the suggestions and made changes, which were highlighted in the re-submitted manuscript. Please see below, for a point-by-point response to the reviewer’s comments and suggestions.

Question: Perhaps you should have considered including a control group of healthy subjects and another group with diabetes mellitus type 2

Answer: Thanks very much for the reviewer's advice. Long standing DM is a risk factor for PDAC, these patients were typically obese and systemic immune-inflammation was considered as a potential cause. However, T3cDM maybe a paraneoplastic phenomenon and the mechanisms have not been elucidated in detail. Thus, in this study we did not analysis another group with T2DM or healthy subjects due to the potential mix effect from T2DM and PDAC. Really, we will plan to carry out research to identify the difference in healthy subjects and T2DM patients with PDAC in the future.

Once again, thank you very much for your comments and suggestions and hope that the correction will meet with approval.

Yours sincerely,

Guanhua Chen

Reviewer 2 Report

The current manuscript analyzes the correlation of systemic-immune inflammation index (SII) and diabetes mellitus secondary to pancreatic cancer. This is an interesting topic; there are however, several concerns.

Diabetes mellitus is both a risk factor as well as a consequence of pancreatic cancer. The authors focus on the latter one, i.e. type 3c diabetes. This should be more clearly stated in the introduction.

Seven patients were excluded because of pre-existing (>1 year) diabetes, whereas 77 patients had newly diagnosed type 3c diabetes. This ratio is somehow unexpected. Can we be sure that all these patients were newly diagnosed? In this line, one would not expect a higher BMI in type 3c diabetes patients.

Could the authors provide information regarding when diabetes was diagnosed in these 77 patients? Were all newly diagnosed (i.e. at hospital admission)?

Long standing diabetes mellitus is a risk factor for pancreatic cancer; these patients are typically obese and most likely display systemic immune-inflammation. In contrast, type 3c diabetes is a paraneoplastic phenomenon and the mechanisms have not been elucidated in detail. In the current analysis, there seems to be a mix of these populations.

“Age and SII were two independently risk factors for diabetes secondary to PDAC”. This is not correct as it suggest a causal relationship. SII was associated with diabetes in this cohort of pancreatic cancer patients.

Author Response

Dear reviewer:

Thank you very much for your comments and suggestions. Your suggestion hit the nail on the head, which made us understand our shortcomings. We accepted the suggestions and made changes, which were highlighted in the re-submitted manuscript. Please see below, for a point-by-point response to the reviewer’s comments and suggestions.

Question 1: Diabetes mellitus is both a risk factor as well as a consequence of pancreatic cancer. The authors focus on the latter one, i.e. type 3c diabetes. This should be more clearly stated in the introduction.

Answer: Thank you very much for the comments, we corrected it in revised manuscript.

Question 2: Seven patients were excluded because of pre-existing (>1 year) diabetes, whereas 77 patients had newly diagnosed type 3c diabetes. This ratio is somehow unexpected. Can we be sure that all these patients were newly diagnosed? In this line, one would not expect a higher BMI in type 3c diabetes patients. Could the authors provide information regarding when diabetes was diagnosed in these 77 patients? Were all newly diagnosed (i.e. at hospital admission)?

Answer:Thanks for the reviewer's questions and suggestions, we have made modifications and supplementary explanations in the revised draft. In fact, there is no precise way to tell whether pancreatic cancer with diabetes is type 2 diabetes or type 3c diabetes. Literature studies have suggested that the definition of new diabetes associated with pancreatic cancer is diabetes occurring 1-2 years prior to tumor diagnosis. In order to more accurately identify pancreatic cancer with diabetes as an emerging disease, our study defined the time period as 12 months. In our study, 77 patients diagnosed with newly diagnosed diabetes, including 17 (22.1%) with the confirmed diagnosis of diabetes mellitus outside the admission with the median time of 5.0 (2.0, 7.0) months. Additionally, the high ration may partially owe to the lack of diabetes education in the region. It is generally believed that high BMI is not only a high risk factor for T2DM, but also a risk factor for PDAC. In addition, patients with PDAC have decreased BMI due to disease, but there is insufficient evidence for changes in BMI in patients with PDAC with DM and PDAC without DM. In our study, the BMI value of the PDAC with DM group was higher than that of the PDAC without group, but it did not reach statistical significance. We have also made corresponding revisions and supplements in the revised draft.

Question 3: Long standing diabetes mellitus is a risk factor for pancreatic cancer; these patients are typically obese and most likely display systemic immune-inflammation. In contrast, type 3c diabetes is a paraneoplastic phenomenon and the mechanisms have not been elucidated in detail. In the current analysis, there seems to be a mix of these populations.

Answer: According to the definition of new-onset diabetes associated with pancreatic cancer, it is difficult to accurately distinguish T2dm from T3cDM, so we strictly defined it as new-onset diabetes 12 months before tumor diagnosis with reference to the literature definition. In addition, we pointed out in the revised draft that the definition was inadequate, which may have mixed up the possibility of partial T2DM.

Question 4: “Age and SII were two independently risk factors for diabetes secondary to PDAC”. This is not correct as it suggests a causal relationship. SII was associated with diabetes in this cohort of pancreatic cancer patients.

Answer: we have corrected the expression in the resubmitted manuscript

Once again, thank you very much for your comments and suggestions and hope that the correction will meet with approval.

Yours sincerely,

Guanhua Chen

Round 2

Reviewer 2 Report

Although some general concerns remain, the authors have satisfactorily answered most of the questions and concerns of the reviewer. I believe that this is an interesting and valid manuscript that has been strengthened by the changes.